# Generative Data Mining with Longtail-Guided Diffusion

**David S. Hayden** [1]   **Mao Ye** [1]   **Timur Garipov** [2]   **Gregory P. Meyer** [1]   **Carl Vondrick** [3]   **Zhao Chen** [4]   **Yuning Chai** [5]
**Eric Wolff** [1]   **Siddhartha S. Srinivasa** [1]

## Abstract

It is difficult to anticipate the myriad challenges that a predictive model will encounter once deployed. Common practice entails a *reactive*, cyclical approach: model deployment, data mining, and retraining. We instead develop a *proactive* longtail discovery process by imagining additional data during training. In particular, we develop general model-based longtail signals, including a differentiable, single forward pass formulation of epistemic uncertainty that does not impact model parameters or predictive performance but can flag rare or hard inputs. We leverage these signals as guidance to generate additional training data from a latent diffusion model in a process we call Longtail Guidance (LTG). Crucially, we can perform LTG without retraining the diffusion model or the predictive model, and we do not need to expose the predictive model to intermediate diffusion states. Data generated by LTG exhibit semantically meaningful variation, yield significant generalization improvements on numerous image classification benchmarks, and can be analyzed by a VLM to proactively discover, textually explain, and address conceptual gaps in a deployed predictive model.

## 1. Introduction

Longtail encounters are common in production ML systems but difficult to anticipate and be robust towards. Recently, the answer has been: add more training data and more compute (Kaplan et al., 2020). However, the time and cost to acquire additional data, especially longtail data, can be prohibitive (e.g. train instances occurs 1 time for every $10^4$ car instances in Berkeley DeepDrive (Yu et al., 2020)). Addi-

[1]Cruise, LLC, San Francisco, CA [2]OpenAI, San Francisco, CA [3]Colombia University, New York, NY [4]Upwork, Palo Alto, CA [5]Meta, Menlo Park, CA. Correspondence to: David Hayden <david.hayden@getcruise.com>.

*Proceedings of the $42^{nd}$ International Conference on Machine Learning*, Vancouver, Canada. PMLR 267, 2025. Copyright 2025 by the author(s).

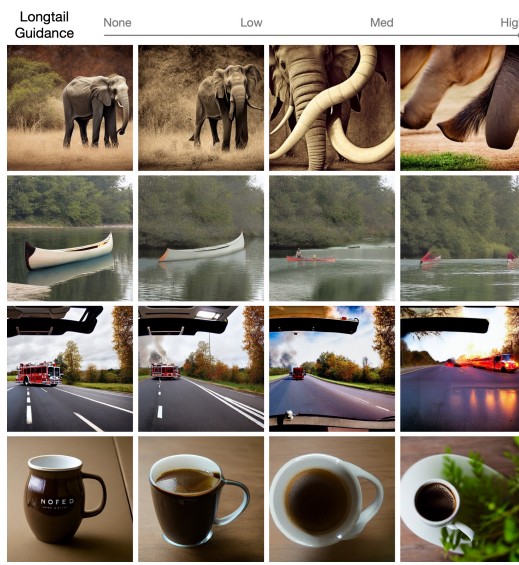

Longtail Guidance    None    Low    Med    High

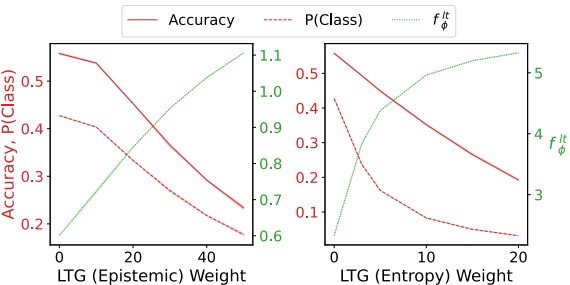

*Figure 1.* Diffusion with Longtail Guidance (LTG) generates synthetic data that are difficult or rare for an existing predictive model. The predictive model can be fine-tuned on this data to improve generalization performance and the the synthetic data can be analyzed to understand conceptual gaps in the predictive model. Guided generations exhibit more extreme views compared to unguided generations. Results in Section 4 use low-to-mid guidance weights.

*Figure 2.* Predictive model performance (left axes, red) and model-based longtail signals $f_\phi^{lt}$ (right axes, green) on synthetic data generated by varying Longtail Guidance weights for model-based longtail signals: Epistemic (left plot) and Entropy (right plot). Guided generations predictably exhibit lower correct class probability, P(Class), lower accuracy, and higher longtail signals, $f_\phi^{lt}$, compared to unguided generations (zero guidance weight), indicating they are more difficult and more longtail from the predictive model's perspective. We ensure that longtail-guided synthetic data generations do not stray out-of-distribution by ensuring P(Class) is lower than unguided generations but well above zero probability.

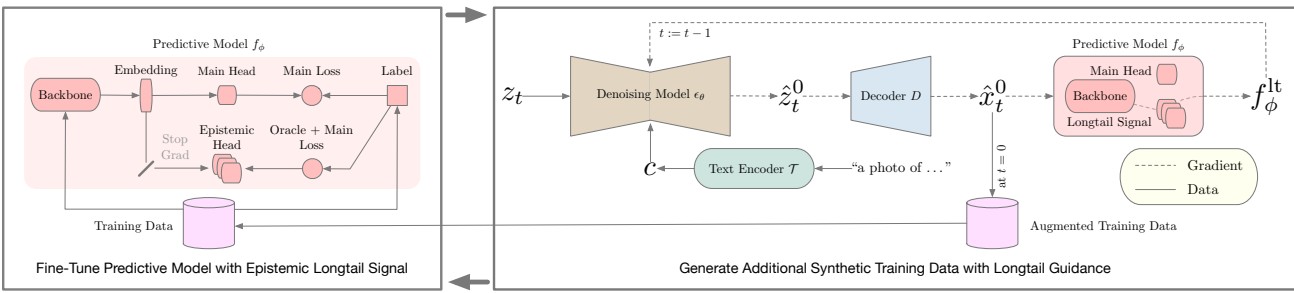

*Figure 3.* In Longtail Guidance, we iteratively fine-tune existing predictive model $f_\phi$ with longtail signal $f_\phi^{\text{lt}}$ (left) then freeze model weights $\phi$ and generate synthetic data with latent diffusion model $\epsilon_\theta$ guided by $f_\phi^{\text{lt}}$ that is, by definition, rare or disproportionately hard for $f_\phi$ (right). Synthetic data are added to $f_\phi$'s training set, and the process repeats.

tional compute can be traded for improved performance; see distillation (Yu et al., 2023b; Gou et al., 2021) and synthetic data generation (Du et al., 2024; Zhang et al., 2023b) for offline approaches, chain of thought (Yao et al., 2024; Wei et al., 2022) for online approaches. While online approaches show promise, they are not applicable to real-time, safety-critical systems such as autonomous driving, which have strict parameter count and inference latency budgets.

Production ML systems are iteratively developed in a continuous, reactive cycle of: collect data, deploy model, encounter longtail edge cases, repeat. We wish to expedite this cycle and move it towards a proactive approach of longtail discovery. Existing offline approaches appeal to larger models (knowledge distillation, synthetic data generation) or summarize training data (data distillation). Yet, they do not directly account for what is rare or difficult for an existing, currently-deployed predictive model.

We wish to trade additional offline compute for performance by generatively mining additional training data that is hard or rare *from the perspective of an existing predictive model*. Distinct from adversarial training and robustness approaches (Bansal & Grover, 2023; Zhang et al., 2022a; Gowal et al., 2021; Hendrycks et al., 2021) which imbue models with invariance to distribution shift or small, often imperceptible changes, we aim to generate semantically meaningful, in-distribution examples that cause high model uncertainty.

Foundation generative models such as Stable Diffusion (Rombach et al., 2022), Pixart (Chen et al., 2025), GPT (Brown et al., 2020; OpenAI, 2023), and Claude (Anthropic, 2024) are exposed to Internet-scale data, but it is not obvious how to use them to generate data that is specifically challenging to an existing predictive model. We seek to explicitly couple the vast knowledge contained in foundation generative models with the specific challenges faced by an *existing* predictive model. But how can we ensure that the generative model produces data that is relevant (has meaningful learning signal) to a given predictive model? Existing approaches, like prompt tuning and reasoning in an embedding space (such as CLIP) (Zhang et al., 2023b;

Du et al., 2024) can generate synthetic data that improves model generalization performance up to a point, but they do not condition on the deployed predictive model.

Following, we develop model-based longtail signals (Section 2), including a lightweight Epistemic Head (Section 2.1), that effectively indicate an example as being hard or rare while keeping the original predictive model intact. We then develop Longtail Guidance (Section 3), a simple method for coupling Internet-scale knowledge in diffusion models with the specific struggles of an existing predictive model. We show that synthetic data produced with Longtail Guidance yields outsized generalization improvements over existing synthetic data generation approaches and that these synthetic data exhibit semantically meaningful variations, including occlusion and extreme views, see Figures 1, 2, 6 and Section 4 for comparisons. In summary, we contribute:

1. Epistemic Head: a differentiable, single forward-pass formulation of epistemic uncertainty that does not impact existing model weights or performance, detects rare or hard examples, and guides data generation towards high-value training examples.

2. Longtail Guidance (LTG): a latent diffusion guidance technique that generates high-value, difficult and/or rare training examples from the perspective of an existing predictive model. It requires no changes in training for the diffusion model or the predictive model.

3. Longtail Introspection: through VLM analysis of LTG-generated data, we find keywords describing what an existing predictive model struggles with. Keywords can be used to prompt diffusion models for high-value data or to inform future real data collection.

## 2. Model-Based Longtail Signals

Longtail is a *suitcase word* (Minsky, 2006) – it does not have a single definition. Two reasonable definitions include:

- **Rare**: An input is longtail if instances similar to it are

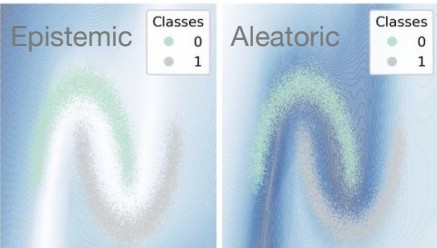

*Figure 4.* Epistemic $E(y, \phi)$ vs aleatoric $\mathbb{E}_\phi[\mathcal{U}(y \mid \phi)]$ uncertainty. Epistemic increases with distance from the data manifold. Aleatoric increases with proximity to the decision boundary.

rare in training data. This definition naturally captures a data-centric view in which events are longtail if their overall occurrence is rare (independent of any model).

- **Hard**: An input is longtail if it is disproportionately difficult for a given model to correctly reason about.

We develop differentiable model-based longtail signals that can flag both rare and hard instances. We then use model-based longtail signals offline to generate additional longtail synthetic training data that provide outsized generalization improvements. Following, we introduce the Epistemic Head, a lightweight addition to any predictive model that does not impact predictive performance but provides superior longtail signals based on detecting hard or rare examples.

We wish to develop longtail signals that are general to loss, architecture (transformer, convolutional), or output space (discrete, continuous). Ideally, they do not affect model performance or require changes to model training. Obvious candidates would be an uncertainty measure of the output, such as entropy or variance. Others can be gathered from the anomaly and out-of-distribution detection literature. One is the Helmholtz free energy, defined as the negative log partition function of an energy model $p(y \mid x)$,

$$E(x) = -T \log \int_{y'} e^{E(x,y')/T} dy' \qquad (1)$$

where $x$ is some input with energy $y$. This energy can be computed in typical classification models as the negative, temperature-scaled log-sum-exp of the logits, $E(x) = -T \log \sum_i e^{f_i(x)/T}$ where $f_i(x)$ is the $i$th logit of a predictive model $f(x)$ that forms a probability distribution by softmax. Energy was shown to be effective in discriminating in-distribution and out-of-distribution data (Liu et al., 2020). Like entropy and variance, it requires no model adjustment or retraining. We use entropy and energy as baselines.

## 2.1. Epistemic Head

Traditional measures of predictive uncertainty, like entropy, do not distinguish what is rare from what is ambiguous

or hard. To address longtail scenarios, we would ideally account for both. To do so, we decompose predictive uncertainty into two components: epistemic and aleatoric (Figure 4). For test input $x$ with unknown label $y$ and predictive model parameters $\phi$ drawn from distribution $\Phi$, epistemic uncertainty is defined as (Depeweg et al., 2018),

$$E(y, \Phi) = \mathcal{U}(y) - \mathbb{E}_\phi [\mathcal{U}(y \mid \phi)] \qquad (2)$$

where we implicitly condition on $x$ in each term, and where $\mathcal{U}$ is an uncertainty measure, such as entropy for discrete $y$ or variance for continuous $y$. If entropy, Eqn 2 is equivalent to measuring the mutual information between parameter distribution $\Phi$ and unknown target $y$, which informs us about how much knowing about one tells us about the other.

In principle, epistemic uncertainty can be computed by sampling from the posterior predictive distribution:

$$p(y \mid x) = \int p(y \mid \phi, x) \, p(\phi \mid x_{\text{train}}, y_{\text{train}}) \, d\phi. \qquad (3)$$

The first term is a likelihood for test label $y$ and the second term is a posterior distribution for model parameters $\phi$, conditioned on all previous training data $(x_{\text{train}}, y_{\text{train}})$.

Performing the inference required to sample from Eqn 3 is intractable for modern neural networks. Nevertheless, it can be approximated. One approach is variational inference, typified by Monte Carlo dropout (Gal & Ghahramani, 2016), where the same input is passed through a network many times (often 50 or more forward passes), with test time variation in each pass enabled by random Bernoulli dropout. Unfortunately, this would be expensive to compute and is not conveniently differentiable due to discrete sampling (Jang et al., 2016). It is also bested in practical terms by a model ensemble, where a small number of models, often $3 - 5$, are trained (independently (Lakshminarayanan et al., 2017) or otherwise (Maddox et al., 2019)). However, computing a gradient (which we will need to generate longtail synthetic data) across $K$ instances of a model introduces substantial memory overhead and compute latency.

In Figure 3 (left), we introduce a lightweight ensembling technique called the Epistemic Head, which provides a superior longtail signal with no impact on model performance and negligible changes to model parameter count, training time, and inference time (see Supplement A.5). Inspired by LoRA (Hu et al., 2021) and the Hydra architecture (Tran et al., 2020), we duplicate the head of an existing prediction model $K$ times and jointly train with the same loss as the base model but with diversity encouraged through an oracle loss (Guzman-Rivera et al., 2012) that only propagates per-example loss through the best-performing head. Head outputs act as fixed-point samples from the posterior predictive (Eqn 3) under a prior determined by weight initialization and permit differentiable computation of longtail signals in a single forward pass, including $E(y, \Phi)$,

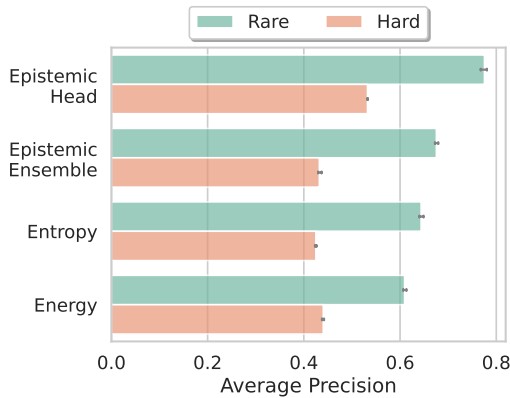

*Figure 5.* The Epistemic Head is a better indicator of rare or hard ImageNet-LT validation examples than entropy, energy, or epistemic signals from an independently-trained ensemble.

$$f_\phi^{\text{lt}}(x) \approx \mathcal{U}\left(\frac{1}{K}\sum_k p(y \mid \phi_k)\right) - \frac{1}{K}\sum_k \mathcal{U}\left(p(y \mid \phi_k)\right). \quad (4)$$

Base model performance is protected by a stop-grad at training time so that, similar to LoRA (Hu et al., 2021), the Epistemic Head does not impact existing model weights. In Figure 5, we demonstrate that longtail signals from the Epistemic Head more effectively indicate rare or hard examples than do entropy or energy. In particular, we train ViT-B ImageNet-LT classifiers according to (Xu et al., 2023) and compare how well our proposed longtail signals (entropy, energy, epistemic) detect rare or hard test examples. Examples are defined as rare if they come from a longtail class. Examples are defined as hard if the model incorrectly predicts the label. In Supplement A.6, we further show that detected examples are disproportionately hard.

## 3. Longtail Guidance

To motivate Longtail Guidance, we briefly review diffusion. Diffusion models learn a continuous data distribution $p_\theta(x_0) = \int p_\theta(x_{0:T})dx_{1:T}$ from a finite set of data samples by defining a forward noising process $q(x_{1:T}|x_0)$ over latent states $x_{1:T}$, and learning a reverse denoising process $p_\theta(x_{0:T})$. In the DDPM (Ho et al., 2020) and DDIM (Song et al., 2020a) formulations, the forward process is a Markov chain that iteratively adds noise according to a schedule $\alpha_{1:T}$ that decreases over steps $1, \ldots, T$,

$$q(x_t|x_{t-1}) = N(\sqrt{\alpha_t}x_{t-1}, (1-\alpha_t)I) \quad (5)$$

However, they differ in the learned reverse process. DDPM models it as a Markov chain,

$$p_\theta(x_t|x_{t-1}) = N(\mu_\theta(x_t, t), \Sigma_\theta(x_t, t)) \quad (6)$$

where, by reparameterization for numerical stability, the diffusion network $\epsilon_\theta(x_t, t)$ learns to predict the previously-

sampled noise at each step rather than the process mean. In DDIM, the reverse process is instead modeled as non-Markovian. At each step, it predicts,

$$x_{t-1} = \sqrt{\alpha_{t-1}}\hat{x}_0^{(t)} + v_t + \sigma_t\epsilon_t \quad (7)$$

for terminal state estimate $\hat{x}_0^{(t)}$, directional vector $v_t$ pointing towards $x_t$, and sampled noise $\epsilon_t \sim N(0, I)$,

$$\hat{x}_0^{(t)} = \alpha_t^{-0.5}\left(x_t - \sqrt{1-\alpha_t}\epsilon_\theta(x_t, t)\right) \quad (8)$$

$$v_t = (1 - \alpha_{t-1} - \sigma_t^2)^{1/2}\epsilon_\theta(x_t, t). \quad (9)$$

SDE formulations of diffusion (Song et al., 2020b) generalize DDPM and DDIM. Denoising models can be learned in one framework (DDPM) and sampled in another (DDIM).

As stated, diffusion models sampled according to Eqns 6, 7 are unconditional; they will sample instances that are distributed approximately according to the data. This can be changed with guidance. In classifier-free guidance (Ho & Salimans, 2022), the diffusion model is trained on pairs $(x, c)$ for data $x$ and conditioning vector $c$ (for example, class labels or CLIP-encoded text). In contrast, classifier guidance (Dhariwal & Nichol, 2021) can be used during the sampling process even when no conditioning information was available at diffusion training time. It operates by biasing the denoising estimate in the direction of the gradient of a differentiable signal, $\nabla f(x_t, t)$,

$$\hat{\epsilon}_t = \epsilon_\theta(x_t, t) - w\nabla_{x_t}f(x_t, t)\sigma_t. \quad (10)$$

Commonly, $f(x_t, t) = \log p(y_i \mid x_t, t)$, the log probability of class $i$ under trained classifier $f_{\phi'}$. But this only works if classifier $f_{\phi'}$ is trained on intermediate noisy diffusion states $x_{1:T}$ and accepts denoising step $t$ (thus, it trains on triples of (noisy data $x_t$, step $t$, and label $y$) instead of a standard formulation of (clean data $x$ and label $y$)). Training $f_{\phi'}$ on intermediate diffusion states is required because the distribution of $x_t$ differ from the original data $x$. Classifier-free guidance can be combined with classifier guidance.

Using classifier guidance to generate synthetic data in the longtail of an existing production model $f_\phi$, as defined by model-based longtail signal $f_\phi^{\text{lt}}(x)$ is appealing because it can be applied to existing diffusion models without re-training. However, as stated, classifier guidance presents a dilemma: we must either fine-tune a noise-aware model $f_{\phi'}(x_t, t)$ from the original model $f_\phi(x)$ on noisy, intermediate diffusion states, at which point $f_{\phi'}$ no longer reflects the production model performance of $f_\phi$, or we must deploy a production model $f_{\phi'}$ that wastes capacity on intermediate diffusion states that it will not encounter in production.

There is an additional challenge with using classifier guidance for longtail data generation: SOTA diffusion models perform training and sampling in a lower dimensional latent space $\mathcal{Z}$ by first encoding the original data using a

Low        Medium        High

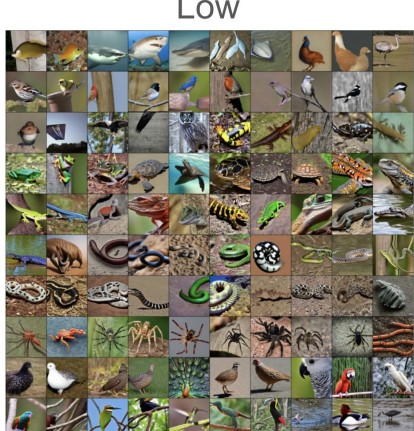 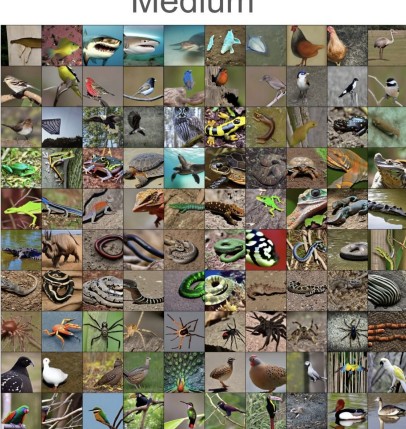 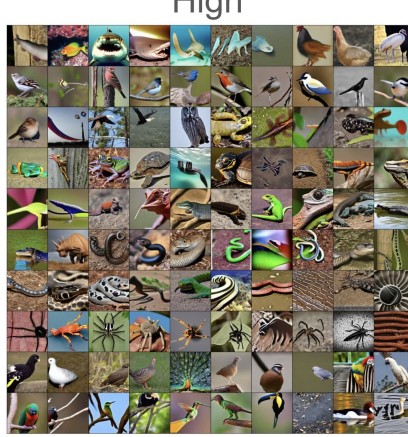

*Figure 6.* Synthetic data generated for 100 ImageNet classes with increasing Longtail Guidance strength, guided by a SOTA ViT ImageNet-LT classifier $f_\phi$. Views frequently become more extreme, occluded, or cut off. They also become more difficult for $f_\phi$ (see Figure 2). Best experiment results use low-to-mid Longtail Guidance weights (see Section 4 for details).

pretrained VAE, noising (at training) and denoising (at inference) among latent states $z_{0:T}$, then decoding the final result back to data space $\mathcal{X}$. How can we pass latent $z_t$ through production model $f_\phi$ when it operates in data space $\mathcal{X}$?

---

**Algorithm 1** Longtail Guidance

**Input:** Latent diffusion model $\epsilon_\theta(z_t, t)$, predictor $f_\phi(x)$ latent decoder $D$, noise schedule $\sigma_{1:T}$, weight $w$

**Initialize:** $z_T \sim \mathcal{N}(0, I)$

**for** $t = T - 1, \ldots, 0$ **do**
    Estimate terminal latent state $\hat{z}_t^0 = P(z_t)$ as in Eqn. 8
    Decode terminal data state: $\hat{x}_t^0 = D(\hat{z}_t^0)$
    Compute model longtail signal $f_\phi^{\text{lt}}(\hat{x}_t^0)$ as in Eqn. 4
    Bias denoising estimate as in Eqn. 11
    Compute $z_{t-1}$ as in Eq. 7
**end**
**return** $x = D(z_0)$

---

With Longtail Guidance, we find a simple diffusion guidance approach that couples longtail signals from an existing production model to the Internet-scale knowledge of a latent diffusion model. Surprisingly, it requires no retraining of either the diffusion model or the production model and, in particular, it does not require the production model to be trained on intermediate diffusion states or time-conditioned inputs, as is common in Classifier Guidance.

The key idea in Longtail Guidance (LTG) is that we can differentiably estimate a terminal latent state $\hat{z}_t^0 = P(z_t)$ with appropriate diffusion samplers (including DDIM), decode to an estimated terminal data state $\hat{x}_t^0 = D(\hat{z}_t^0)$, compute longtail signal $f_\phi^{\text{lt}}(\hat{x}_t^0)$ from the existing production model (that has only ever seen real production data), and then bias the denoising estimate (in latent space) in the direction of

higher production model longtail signal (See Figure 3 and Algorithm 1 for complete details):

$$\hat{\epsilon}_t = \epsilon_\theta(z_t, t) - w\nabla_{z_t} f_\phi^{\text{lt}}(D(P(z_t)))\,\sigma_t. \quad (11)$$

It is unintuitive that LTG would work since production model $f_\phi$ has only ever trained on clean (data, label) pairs, not intermediate diffusion states. Empirically, however, we find that we can generate synthetic data for which production model $f_\phi$ exhibits lower probability of the correct class, lower accuracy, and higher longtail signal. And so long as we do not adjust the longtail guidance weight $w$ too high, the diffusion model reliably generates data that adheres to the expected class label.

Figures 1, 6 and Supplement A.9 show example longtail synthetic ImageNet data using a SOTA ViT classifier (Xu et al., 2023) for guidance. Figure 2 quantitatively analyzes model performance and longtail signals on LTG-guided synthetic data generation over multiple runs of $24k$ generations spanning all ImageNet classes. Notably, the probability of the expected class can be decreased to one-third of its original value (from the perspective of production model $f_\phi$), while staying in-distribution (as evidenced by generalization improvements in Section 4, see also Section 3.1) and the model longtail signal can be more than doubled over baseline (unguided) synthetic data generation. Qualitatively, LTG data exhibit semantically meaningful changes, including more extreme, occluded, or cut-off views.

Longtail Guidance is similar to Universal Guidance (Bansal et al., 2023) in that both perform differentiable decoding of latent states (see also (Jiang et al., 2023; Song et al., 2023; Dou & Song, 2024)). However, Universal Guidance is significantly more expensive and fails to remain in-distribution when guided by model-based longtail signals. In particular, Universal Guidance performs so-called recurrent and back-

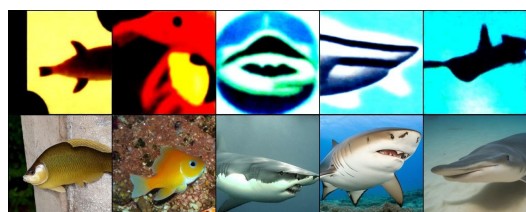

*Figure 7.* Universal Guidance (top) vs Longtail Guidance (bottom). Universal Guidance, when driven by model-based longtail signals, successfully raises those signals but it does so by generating data that is no longer in-distribution. It is also much more expensive.

ward sampling passes, where recurrent sampling permits each diffusion step to refine its denoising estimate (each time computing a costly gradient) and each backward pass optimizes an objective (within each denoising and recurrent iterate, itself requiring multiple optimization iterates). See Figure 7 for direct comparison.

Although Longtail Guidance is not the first approach to use the predicted, decoded terminal state $D(P(x_t))$, we uniquely (to our knowledge) show in Supplement A.7 why the predictive model performing guidance does not need to be trained on intermediate diffusion states: $D(P(x_t))$ is closer in distribution to real training data than is $x_t$.

LTG requires no training for either the diffusion or the production model. Each can be used off-the-shelf. It's primary limitation is that the gradient $d\hat{x}_t^0/d\hat{z}_t^0$ calculated by decoding is expensive (see Supplement A.5).

### 3.1. Remaining In-Distribution

We ensure that LTG-generated data remain in-distribution but longtail by choosing a guidance weight such that (1) the probability of the desired class is lower than baseline unguided synthetic data generation but nonzero, and (2) that the model-based longtail signal $f_\phi^{\text{lt}}$ evaluates higher than it does for baseline unguided synthetic data. Figure 2 shows that both are achievable; in fact, we find that a single weight can be chosen one time and used for all future generations (across classes, datasets and fine-tuning epochs). With this approach, we find that FID (Heusel et al., 2017) scores are lower and generative precision+recall (Kynkäänniemi et al., 2019) are higher when comparing LTG-generated synthetic data to real data (within-classes) than when comparing real data to itself (between classes).

## 4. Experiments

In Section 4.1, we compare predictive model generalization improvements when training data is augmented with synthetic data generated by Longtail Guidance, when training data is augmented with existing synthetic data generation approaches, and when training data is augmented with tradi-

tional data augmentations approaches. In Section 4.2, we demonstrate that LTG improves generalization performance of SOTA ViT models that compensate for class-imbalance, particularly for non-synthetic longtail data. In Section 4.3, we reduce synthetic data generated with Longtail Guidance to a set of text descriptions that describe attributes of a predictive model's longtail. We demonstrate that these descriptions are meaningful by showing that they produce higher-value synthetic data than manually prompt-tuned diffusion (in terms of predictive model generalization performance). In Section 4.4, we examine why Longtail Guidance outperforms existing synthetic data generation approaches.

### 4.1. LTG Improves Predictive Model Generalization

We compare Longtail Guidance with three measures (Epistemic, Entropy, Energy) to the recent Guided Imagination Framework (GIF) (Zhang et al., 2023b) and Dream the Impossible (Dream-ID) (Du et al., 2024) synthetic data generation approaches. For baselines, we compare to prompt-tuned Stable Diffusion 1.4 (SD), prompt-tuned DALL-E2, and masked autoencoder (MAE) data generation (He et al., 2022). We also compare to data augmentation baselines: Cutout (DeVries, 2017), GridMask (Chen et al., 2020), RandAugment (Cubuk et al., 2020), AutoAugment (Cubuk et al., 2019), CutMix (Yun et al., 2019), AugMix (Hendrycks et al., 2019) and adversarial robustness approaches DeepAugment (Hendrycks et al., 2021), and MEMO (Zhang et al., 2022a). Finally, we report CLIP zero-shot and distillation performance (Radford et al., 2021).

We evaluate on seven natural image datasets spanning fine-grained, coarse-grained, and mixed coarse/fine classification tasks. Datasets include ImageNet, ImageNet-V2, ImageNet-A, Stanford Cars (Krause et al., 2013), Oxford Flowers (Nilsback & Zisserman, 2008), Oxford Pets (Parkhi et al., 2012), and Caltech101 (Fei-Fei et al., 2004). We reproduce baseline predictive models (Original) that are not exposed to synthetic data, with comparable performance (within 1%) of the baselines used in (Du et al., 2024; Zhang et al., 2023b).[1] We iteratively perform additional fine-tuning with synthetic training data that is equal in quantity across all conditions, use the same augmentations (random rotation, reflection, and center cropping), and report generalization performance as top-1 accuracy. Results are averaged over three runs. Additional experiment details are in Supplement A.1 and dataset details are in Supplement A.3. We fix hyperparameters (LTG weight, number of Epistemic Heads) across all datasets according to Supplement A.8.

In Table 1, we compare to baselines under the Dataset Expansion task as defined in GIF (Zhang et al., 2023b). For parity with the architecture and quantity of synthetic data

---

[1] Baseline model checkpoints were not available so we reproduce them using the reported architecture and training recipe.

used by GIF, we fine-tune ResNet-50 models for 100 epochs and generate synthetic data equivalent to $20\times$ the original dataset size for Caltech, Cars and Flowers, and $30\times$ the original dataset size for Pets. GIF generates all synthetic data at once whereas we evenly distribute data generation throughout training epochs to better capture evolving model-based longtail challenges, see Supplement A.1 for details. In each expansion, synthetic examples per class are equal in quantity to the original number of real examples.

In Table 2, we compare to baselines under the in-distribution synthetic data generation task as defined in Dream the Impossible (Dream-ID) (Du et al., 2024) and discussed in Supplement A.1. For parity with the architecture and quantity of synthetic data used by Dream-ID, we train ResNet-34 models for 100 epochs and generate 1k synthetic data per class. Models are trained on ImageNet-100 and evaluated on the same ImageNet-100, ImageNet-A, and ImageNet-V2 subsets as defined in Dream-ID and restated in Supplement A.1. In each benchmark, Longtail Guidance provides substantial improvements over existing synthetic data generation methods and data augmentation baselines. In particular, it provides an average 5.6 points of additional top-1 accuracy compared to GIF or Dream-ID, and an average 15.0 points of additional top-1 accuracy compared to text-prompted Stable Diffusion. Aggregate improvements from the Epistemic Head over alternative model-based longtail signals (energy, entropy) are mild but significant and come at negligible inference or training time cost as described in Supplement A.5. In Supplement A.4, we show that predictive model generalization can be further improved by engaging in more cycles of fine-tuning and synthetic data generation.

Strikingly, we find that predictive model generalization when fine-tuned on LTG-generated synthetic data eclipses generalization when fine-tuned on GIF-generated data at substantially lower synthetic data volumes: 20% for Caltech, 25% for Cars, 30% for Flowers, and 23% for Pets, for an overall average of 24.5%. Thus, LTG-generated data not only provide higher overall generalization improvements, but they do so at more than $4\times$ the synthetic data efficiency. Furthermore, LTG-generated data provide value in both high synthetic data volume regimes ($20 - 30\times$ original dataset size for GIF comparison) and low synthetic data volume regimes (less than $1\times$ original dataset size for Dream-ID and ImageNet-LT comparisons).

### 4.2. LTG Improves Longtail Performance

In Table 3, we train SOTA ViT-based models (LiVT) from scratch on ImageNet-LT according to the longtail-compensation approach of (Xu et al., 2023). In summary, training includes MAE pretraining followed by 100 epochs of BCE loss with a logit adjustment to account for class imbalance. Initial LiVT training is on real data only and uses

*Table 1.* Top-1 Accuracy on coarse-grained (Caltech) and fine-grained (Pets, Cars, Flowers) natural image datasets under the Dataset Expansion task as defined in (Zhang et al., 2023b).

| Dataset | Caltech | Cars | Flowers | Pets | Avg |
|---|---|---|---|---|---|
| Original | 26.3 | 19.8 | 74.1 | 6.8 | 31.8 |
| CLIP | 82.1 | 55.8 | 71.2 | 85.4 | 72.3 |
| CLIP Distill | 33.2 | 18.9 | 75.1 | 11.1 | 34.6 |
| **Expanded** | | | | | |
| Cutout | 51.5 | 25.8 | 77.8 | 38.7 | 48.5 |
| GridMask | 51.6 | 28.4 | 80.7 | 37.6 | 49.6 |
| RandAugment | 57.8 | 43.2 | 83.8 | 48.0 | 58.2 |
| MAE | 50.6 | 25.9 | 76.3 | 39.9 | 48.2 |
| DALL-E2 | 61.3 | 48.3 | 84.1 | 61.7 | 63.8 |
| SD | 51.1 | 51.7 | 78.8 | 57.9 | 59.9 |
| GIF-MAE | 58.4 | 44.5 | 84.4 | 52.4 | 59.9 |
| GIF-DALLE | 63.0 | 53.1 | 88.2 | 66.4 | 67.7 |
| GIF-SD | 65.1 | 75.7 | 88.3 | 73.4 | 75.6 |
| LTG (Energy) | 70.4 | **85.3** | 90.3 | 80.0 | 81.5 |
| LTG (Entropy) | 70.6 | 84.6 | 89.9 | 81.6 | 81.7 |
| LTG (Epistemic) | **71.5** | 85.1 | **90.9** | **82.0** | **82.4** |

*Table 2.* Top-1 Accuracy on the in-distribution synthetic data generation task for ImageNet variants as defined in (Du et al., 2024)

| Methods | ImageNet | ImageNet-A | ImageNet-v2 | Avg |
|---|---|---|---|---|
| Original | 87.28 | 8.69 | 77.80 | 57.92 |
| RandAugment | 88.10 | 11.39 | 78.90 | 59.46 |
| CutMix | 87.98 | 9.67 | 79.70 | 59.12 |
| AutoAugment | 88.00 | 10.85 | 79.70 | 59.52 |
| AugMix | 87.74 | 10.96 | 79.20 | 59.30 |
| DeepAugment | 86.86 | 10.79 | 78.30 | 58.65 |
| MEMO | 88.00 | 10.85 | 78.60 | 59.15 |
| SD | 87.74 | 11.18 | 79.20 | 59.37 |
| DREAM-ID | 88.46 | 12.13 | 80.40 | 60.33 |
| LTG (Energy) | 90.00 | 20.20 | 81.05 | 63.75 |
| LTG (Entropy) | 90.04 | **22.70** | 80.66 | 64.47 |
| LTG (Epistemic) | **90.30** | 22.09 | **81.54** | **64.64** |

many data augmentations including RandAugment, Mixup, and CutMix. Our baseline predictive model, LiVT, nearly matches the generalization performance (top-1 accuracy) of what is reported in (Xu et al., 2023) (LiVT Reproduced: $60.6$ vs LiVT Reported: $60.9$). We then fine-tune LiVT on $24k$ additional synthetic data evenly distributed across all $1k$ classes for 100 epochs with $1e - 4$ learning rate. We use the same Stable Diffusion 1.4 baseline and sampling details as in Section 4.1 (LiVT SD), and also compare Longtail Guidance with three types of model-based longtail signals (LiVT LTG {Energy, Entropy, Epistemic}). Overall generalization improvements on the balanced validation set are mild but significant ($+1.3$ top-1 accuracy). However, performance for longtail classes (Few) improves by four points for baseline diffusion ($+10\%$) but a marked 10 points for Longtail Guidance ($+24\%$)!

Synthetic data always improves Medium and Few performance, which are most impacted by a fixed number of

data generations per class. This experiment demonstrates that Longtail Guidance works with multiple architectures (ResNet, ViT), multiple losses (CE, logit-adjusted BCE), and that it disproportionately improves longtail performance. It also demonstrates that LTG can be composed with finely-crafted training recipes to further improve predictive model generalization performance.

*Table 3.* Top-1 Accuracy on ImageNet-LT by dataset split (Many, Med, Few) and overall (Acc). The validation set is balanced between all 1000 classes but training is highly imbalanced: Few classes have $5 - 19$ examples, Medium classes have $20 - 99$ examples, Many classes have $100 - 1200$ examples.

| Method | Many | Med. | Few | Acc |
|---|---|---|---|---|
| CE (Cui et al., 2019) | 64.0 | 33.8 | 5.8 | 41.6 |
| LDAM (Cao et al., 2019) | 60.4 | 46.9 | 30.7 | 49.8 |
| c-RT (Kang et al., 2019) | 61.8 | 46.2 | 27.3 | 49.6 |
| $\tau$-Norm (Kang et al., 2019) | 59.1 | 46.9 | 30.7 | 49.4 |
| Causal (Tang et al., 2020) | 62.7 | 48.8 | 31.6 | 51.8 |
| Logit Adj. (Wang et al., 2020) | 61.1 | 47.5 | 27.6 | 50.1 |
| RIDE(4E) (Wang et al., 2020) | 68.3 | 53.5 | 35.9 | 56.8 |
| MiSLAS (Zhong et al., 2021) | 62.9 | 50.7 | 34.3 | 52.7 |
| DisAlign (Zhang et al., 2021) | 61.3 | 52.2 | 31.4 | 52.9 |
| ACE (Cai et al., 2021) | 71.7 | 54.6 | 23.5 | 56.6 |
| PaCo (Cui et al., 2021) | 68.0 | 56.4 | 37.2 | 58.2 |
| TADE (Zhang et al., 2022b) | 66.5 | 57.0 | 43.5 | 58.8 |
| TSC (Li et al., 2022e) | 63.5 | 49.7 | 30.4 | 52.4 |
| GCL (Li et al., 2022d) | 63.0 | 52.7 | 37.1 | 54.5 |
| TLC (Li et al., 2022a) | 68.9 | 55.7 | 40.8 | 55.1 |
| BCL (Zhu et al., 2022) | 67.6 | 54.6 | 36.6 | 57.2 |
| NCL (Li et al., 2022c) | 67.3 | 55.4 | 39.0 | 57.7 |
| SAFA (Hong et al., 2022) | 63.8 | 49.9 | 33.4 | 53.1 |
| DOC (Wang et al., 2022) | 65.1 | 52.8 | 34.2 | 55.0 |
| DLSA (Xu et al., 2022) | 67.8 | 54.5 | 38.8 | 57.5 |
| ViT (Dosovitskiy, 2020) | 50.5 | 23.5 | 6.9 | 31.6 |
| MAE (He et al., 2022) | **74.7** | 48.2 | 19.4 | 54.5 |
| DeiT (Touvron et al., 2022) | 70.4 | 40.9 | 12.8 | 48.4 |
| LiVT (Xu et al., 2023) | 72.7 | 56.6 | 40.4 | 60.6 |
| LiVT SD | 71.8 | **58.1** | 44.3 | 61.5 |
| LiVT LTG (Energy) | 70.5 | 57.9 | **50.0** | 61.7 |
| LiVT LTG (Entropy) | 70.9 | 57.9 | 49.8 | 61.8 |
| LiVT LTG (Epistemic) | 71.4 | 57.7 | **50.0** | **61.9** |

### 4.3. LTG Produces Meaningful Longtail Data

We examine synthetic data generated by LTG to determine whether they exhibit meaningful variation from baseline synthetic data generated by prompt-tuned diffusion. This is a difficult and fundamentally qualitative task to manually perform at scale. To make it quantitative, we ask: do text descriptions of LTG-generated data lead to predictive model generalization improvements when they are used to generate additional synthetic data (without LTG)?

Starting with the strongest baseline model $f_\phi$ trained on Flowers (Original from Table 1), we ask a VLM (Liu et al., 2024) to caption real training instances $x_{\text{real}}$ and also syn-

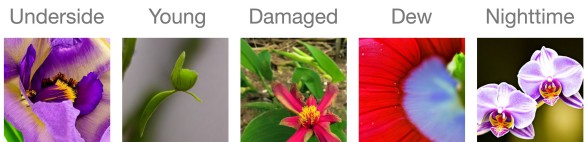

*Figure 8.* Longtail keywords from LTG-generated Flowers data.

thetic data $x_{\text{ltg}}$ generated by LTG as guided by $f_\phi^{\text{lt}}$. For each synthetic instance, $N$ novel keywords are found by computing the token embeddings that are furthest in cosine similarity from all real data token embeddings. An LLM (OpenAI, 2023) is then provided with a set of (caption, novel keyword) pairs for each class and asked to summarize them into $P$ refined prompts. We call this Longtail Introspection.

Refined prompts are used to generate additional synthetic data $x_{\text{inspect}}$ by diffusion (without LTG). We train $f_\phi$ on synthetic data generated by Longtail Introspection and compare it to synthetic data generated by manual prompt tuning (40 unique prompts per class, as in (Zhang et al., 2023b); details in Supplement A.2). Table 4 shows that Longtail Introspection significantly outperforms manually prompt-tuned diffusion (defined in Supplement A.1), quantitatively supporting that data generated by LTG exhibit meaningful variation. Figure 8 shows example longtail keywords.

*Table 4.* Synthetic data generated by prompts based on VLM descriptions of LTG data outperform synthetic data generated by manually prompt-tuned diffusion, quantitatively suggesting that data generated by LTG exhibit meaningful and challenging variation from the perspective of predictive model $f_\phi$.

| Condition | Accuracy |
|---|---|
| No Synthetic Data | 74.1 |
| Manual Prompt Tuning | 78.8 |
| Longtail Introspection | **84.3** |

### 4.4. Discussion

Longtail Guidance significantly outperforms leading synthetic data generation baselines including GIF and Dream-ID even though it requires no prompt tuning. To explain this, we note that GIF and Dream-ID reason in latent CLIP space to create new embedding vectors with which to prompt a diffusion model. For each real data instance, GIF creates $K$ additional synthetic instances by jointly optimizing $K$ embedding vectors (initialized by CLIP image encoding) such that they are difficult for CLIP zero-shot classification (high entropy), remain close to the target class (high log prob of target class), and diverse (high KL divergence to a mean embedding). In Dream-ID, a separate model is trained to predict class embedding vectors from (real data, CLIP-based class text embedding vector) pairs. From this, a manifold in CLIP space can be sampled along class boundaries to generate new diffusion conditioning vectors.

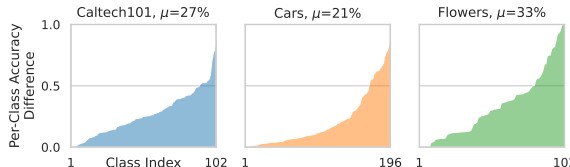

*Figure 9.* Reasoning in latent CLIP space for synthetic data generation does not account for differences in what is difficult for CLIP versus what is difficult for a production model. We plot (sorted) absolute difference in per-class accuracy between a CLIP zeroshot classifier and a well-trained production model. Even when aggregate performance is similar, as in Flowers (74.0% vs 71.2% top-1 accuracy), average per-class difference is high (33%).

We posit that synthesizing data that is difficult or rare for a foundation model, like CLIP, is different from synthesizing data that is difficult or rare for a specific, deployed predictive model. Figure 9 quantifies this by showing that, even when CLIP zero-shot performance and a production model have similar aggregate performance 74.1% vs 71.2%, the classes (and data instances) that are difficult for CLIP are different from the classes that are difficult for the production model–with average per-class accuracy differences of 34%!

In both GIF and Dream-ID, reasoning is only conditioned on training data and restricted to CLIP embedding space. In Longtail Guidance, we instead condition on longtail signals coming directly from the predictive model we wish to improve. We know that these signals identify rare or disproportionately hard examples (Figure 5) and that synthetic data generated by LTG reliably have lower probability of correct classification, lower accuracy, and higher longtail signals (Figure 2) under the predictive model. This is most strikingly demonstrated in the more than 80% generalization improvement over Dream-ID on the naturally challenging examples of ImageNet-A.

## 5. Related Work

Diffusion has seen wide success, particularly in image generation, where it outperform GANs in image quality and diversity without suffering from unstable training or mode collapse (Dhariwal & Nichol, 2021). Recent work has seen progress in handling large data dimensions with latent spaces (Chen et al., 2025; Podell et al., 2023; Rombach et al., 2022) or hourglass networks (Crowson et al., 2024), improved sampling (Lu et al., 2022; Karras et al., 2022; Ho et al., 2020; Song et al., 2020a), additional data domains (Ran et al., 2024; Pronovost et al., 2023; Zhong et al., 2023), and personalization (Ruiz et al., 2023; Kumari et al., 2023). Much work has also been done on new forms of guidance (Wallace et al., 2023b; Yu et al., 2023a; Wallace et al., 2023a; Zhang et al., 2023a) beyond just classifier guidance and classifier-free guidance (Dhariwal & Nichol,

2021; Ho & Salimans, 2022). Universal Guidance (Bansal et al., 2023) and Diffusion Posterior Sampling (Dou & Song, 2024) are most relevant and are discussed in Section 3.

Synthetic training data from generative models has been considered since GANs (Li et al., 2022b), but has started to come of age with diffusion (Azizi et al., 2023; Zhou et al., 2023), particularly for high-resolution datasets where fidelity matters. GIF (Zhang et al., 2023b) and Dream-ID (Du et al., 2024), discussed in Section 4.4, are most relevant.

Model signals have been used for out-of-distribution or adversarial detection (Huang et al., 2021; Hsu et al., 2020; Cohen et al., 2020), particularly model uncertainty (Van Amersfoort et al., 2020) or feature density (Lee et al., 2018). Epistemic uncertainty has been developed in expensive Bayesian (Gal & Ghahramani, 2016) or model ensemble contexts (Liu et al., 2019a; Depeweg et al., 2018; Wilson & Izmailov, 2020; Lakshminarayanan et al., 2017), though, to our knowledge, not in a single, differentiable forward pass as we have done. Work on longtail robustness has primarily focused on addressing a-priori known class imbalance. Datasets include ImageNet-LT, Places-LT, (Liu et al., 2019b) and iNaturalist (Van Horn et al., 2018). Mitigations include pretraining (He et al., 2022; Bao et al., 2021), distillation (Xiang et al., 2020), reweighted loss (Xu et al., 2023; Ross & Dollár, 2017), selective real data mining (Jiang et al., 2022), or low-density data sampling (Um & Ye, 2024).

## Conclusion

We develop model-based longtail signals that do not impact model weights or performance and are leveraged by Longtail Guidance, a synthetic data generation approach that explicitly conditions diffusion on an existing predictive model to generate examples that are rare or hard from that model's perspective. We show that training on LTG-generated data provides strong, data-efficient generalization improvements across eight datasets. We further demonstrate that longtail synthetic data generations can be rendered into meaningful text descriptions and keywords that can aid future (real or synthetic) data collection priorities.

Predictive models are being deployed more than ever before. Increasingly, they will encounter longtail scenarios that human operators cannot easily predict in advance. Foundation models can be used to mitigate some risk, but we cannot shoehorn the entirety of Internet-scale knowledge into every deployed model due to capacity and compute constraints. By letting an existing predictive model *speak for itself*, we can expend offline compute to generatively mine high-value synthetic training data. Further reducing these synthetic examples to human- and machine-readable text suggests a future where we can move away from slow, reactive longtail mitigation towards fast, proactive longtail discovery.

## Impact Statement

We develop model-based longtail signals, model-conditioned longtail synthetic data generation, and longtail interpretation techniques that have the potential to discover and mitigate production model failure modes before they occur. This has tremendous potential for improving the quality and safety of deployed ML applications. However, they may also have negative impacts if human operators rely too heavily on our approaches and not enough on traditional safeguards. We recommend our methods be used to supplement, not replace, standard deployment practices; most importantly, that regular and satisfactory evaluation on non-synthetic evaluation data be used as a requirement for model deployment.

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

# A. Appendix

## A.1. Comparison to GIF and Dream-ID

In the main paper, Table 1, we compare to GIF, the Guided Imagination Framework (Zhang et al., 2023b). We use the same predictive model architecture (ResNet50 trained from scratch), generate the same quantity of synthetic data ($30\times$ dataset size for Pets, $20\times$ dataset size for Caltech101, Cars, and Flowers), the same diffusion model (Stable Diffusion v1.4), the same diffusion sampler (DDIM), and the same number of sampling iterates (50). We reproduce baseline models (not exposed to synthetic data or data augmentation conditions) to match the performance of GIF baselines (Original row in Table 1) to within $1\%$ of the generalization performance reported in GIF. We then fine-tune for 100 epochs, generating synthetic data with Longtail Guidance according to the schedule in Table 5 (e.g. generate synthetic data in epoch 0, fine-tune until epoch 5, generate more synthetic data, fine-tune until epoch 10, . . . ). We experimentally found that synthetic data iteratively generated throughout fine-tuning outperformed synthetic data generated all at once and speculate that this is due to the model-based longtail evolving throughout training. We also experimentally found that using total uncertainty (epistemic + aleatoric, the first term in the RHS of Equation 4) as the LTG guidance signal from the Epistemic Head was overall slightly more performant for downstream predictive model generalization improvements (as compared to epistemic or aleatoric alone), likely because several of our benchmark datasets involve fine-grained distinctions that benefit from additional synthetic data, not just towards the edge of the class manifold (epistemic) but also along decision boundaries (aleatoric). See Figure 4 for visualization.

Synthetic data are distributed across classes according to their frequency in the original training data. Fine-tuning is with the Adam optimizer, cosine annealing learning rate schedule, and $1e-3$ learning rate. As in GIF, we train with random rotations ($\pm15°$), $224 \times 224$ crops, and horizontal flips.

In the main paper, Table 2, we compare to the in-distribution synthetic data generation task (Dream-ID) from Dream the Impossible (Du et al., 2024). As in Dream-ID, we start with ImageNet-pretrained ResNet-34 models and fine-tune for 100 epochs with the same training details as above, except that we match the synthetic data quantity used in Dream-ID by generating a total of 1000 images per class, evenly distributed every 5 epochs (50 images per class per 5 epochs).

Different from GIF or Dream-ID is that we build synthetic data over the course of model fine-tuning, to better capture evolving model-based longtail challenges (whereas GIF, Dream-ID, and other synthetic data baselines generate all synthetic data at once since they are not model-conditioned). Also different is that LTG does not need prompt tuning to force synthetic data diversity. LTG text prompts are generic: "a photo of `Class`" whereas GIF prompts (and diffusion baselines) are, "a `Noun` `Adjective` of `Class`" where `Noun` is randomly sampled to be one of ("image", "oil painting", "cartoon image", "sketch", "pencil sketch") and `Adjective` is randomly sampled to be one of ("", "colorful", "stylized", "bright", "sheared", "solarized", "posterized", "high-contrast"), for a total of 40 unique prompts per class. In LTG, diversity is automatic based on the production model's evolving longtail signals.

All experiments are performed on 8xH100.

| Dataset | Total Expansion Ratio | Synthetic Data Generation Epochs |
|---|---|---|
| Pets | $30\times$ | $0, 5, \ldots, 45, 50; 52, 54, \ldots, 96, 98$ |
| Caltech101, Cars, Flowers | $20\times$ | $0, 5, \ldots, 90, 95$ |

*Table 5.* Synthetic Data Generation Schedule by Dataset

## A.2. Longtail Introspection

We construct refined prompts for each class by prompting LLaVA-1.6 7B (Liu et al., 2024) to generate a description for two sets of synthetic images: 100 generated by Longtail Guidance and 100 generated by baseline diffusion. VLM prompts are of the form: *This is an image of <Class>. Describe it in detail*. For each output VLM description, we compute per-token BERT embeddings (Devlin, 2018). This provides us with two token embedding distributions: longtail and baseline.

For each token embedding in the longtail distribution, we compute cosine similarity to the nearest token embedding in the baseline distribution. For each longtail image, we define the $K$ tokens that are furthest from the the baseline distribution as longtail keywords. We retain the (description, keyword) pairs of the $P = 40$ longtail examples per class whose keyword tokens are furthest from the base distribution.

Following, we create $P = 40$ refined prompts per class by prompting GPT-4o (OpenAI, 2023) with: *<VLM description of the image> The following keywords describe the key features of the description above: <Keyword 1>, <Keyword 2> ... Use a complete sentence to summarize the key features. The sentence should start with: A photo of <Class> that....*

Example (image, keyword) pairs are displayed in Figure 8 of the main paper. Example refined prompts are displayed in Table 6. We emphasize that not all keywords found by this method are immediately meaningful (e.g. satin as in *satin-like pedals*, ether as in *ethereal appearance*, but they can be quickly scanned for longtail themes and, quantitatively, the process improves generalization performance beyond manual prompt tuning.

---

A photo of a bearded iris that showcases its deep purple petals with a yellow and brown pattern running through the **underside**, set against a blurred background to emphasize the flower's texture and color.

A photo of sweet pea that showcases a **young** plant with a single, unfurled green leaf and a small, green flower bud beginning to open, set against a softly blurred green background, highlighting the delicate texture and promise of the flower to come.

A photo of red ginger that showcases a vibrant red flower with a yellow center, surrounded by slightly **damaged** green leaves, set against a blurred natural background.

A photo of morning glory that showcases vibrant petals with a gradient from deep red to blue-purple, arranged in a spiral pattern, with smooth texture and **dew** droplets, set against a blurred background to emphasize the flower's intricate details and colors.

---

*Table 6.* Example refined prompts generated by Longtail Introspection for Flowers data. Keywords are **bolded**.

## A.3. Dataset Details

We summarize each dataset used in Table 7. Additional details can be found in the (Zhang et al., 2023b) for Caltech, Cars, Pets, and Flowers or (Du et al., 2024) for ImageNet-100, ImageNet-A, and ImageNet-V2 variants. Notably, (Du et al., 2024) creates evaluation subsets of ImageNet-A and ImageNet-V2 that overlap with the training classes of ImageNet-100, listed in Figure 10. For fair comparison, we also train on ImageNet-100 and evaluate on the ImageNet-100 (Eval), ImageNet-A, and ImageNet-V2 subsets as defined in (Du et al., 2024). We emphasize that these datasets exhibit example counts that are common for longtail data in production settings.

| Dataset | Classes | #Train | #Val | #Synthetic |
|---|---|---|---|---|
| Caltech101 | 102 | 3060 | 6084 | 61200 |
| Cars | 196 | 8144 | 8041 | 162880 |
| Flowers | 102 | 6557 | 1632 | 131140 |
| Pets | 37 | 3680 | 3669 | 110400 |
| ImageNet-100 | 100 | 129860 | 5000 | 100000 |
| ImageNet-A* | 41 | - | 1852 | - |
| ImageNet-V2* | 100 | - | 10000 | - |
| ImageNet-LT | 1000 | 115846 | 20000 | 24000 |

*Table 7.* Overview of datasets with number of classes, training samples, validation samples, and synthetic samples. *ImageNet-A and ImageNet-V2 are not trained on; they are only used for evaluation.

## A.4. Can We Continuously Mine for Additional High-Value Synthetic Data?

| Dataset | Caltech | Cars | Flowers | Pets | Avg |
|---|---|---|---|---|---|
| LTG (100 epoch) | 71.5 | 85.1 | 90.9 | 82.0 | 82.4 |
| LTG (200 epoch) | 72.9 | 85.4 | 92.6 | 82.1 | 83.3 |
| LTG (300 epoch) | 72.9 | 85.4 | 92.6 | 82.2 | 83.3 |

*Table 8.* Top-1 Accuracy when training with Longtail Guidance for many epochs.

In Table 8 we ask: if Longtail Guidance generates high-value training data for predictive model $f_\phi$, can we continuously

```
n01498041 n01514859 n01582220 n01608432 n01616318 n01687978 n01776313 n01806567 n01833805 n01882714
n01910747 n01944390 n01985128 n02007558 n02071294 n02085620 n02114855 n02123045 n02128385 n02129165
n02129604 n02165456 n02190166 n02219486 n02226429 n02279972 n02317335 n02326432 n02342885 n02363005
n02391049 n02395406 n02403003 n02422699 n02442845 n02444819 n02480855 n02510455 n02640242 n02672831
n02687172 n02701002 n02730930 n02769748 n02782093 n02787622 n02793495 n02799071 n02802426 n02814860
n02840245 n02906734 n02948072 n02980441 n02999410 n03014705 n03028079 n03032252 n03125729 n03160309
n03179701 n03220513 n03249569 n03291819 n03384352 n03388043 n03450230 n03481172 n03594734 n03594945
n03627232 n03642806 n03649909 n03661043 n03676483 n03724870 n03733281 n03759954 n03761084 n03773504
n03804744 n03916031 n03938244 n04004767 n04026417 n04090263 n04133789 n04153751 n04296562 n04330267
n04371774 n04404412 n04465501 n04485082 n04507155 n04536866 n04579432 n04606251 n07714990 n07745940
```

*Figure 10.* List of ImageNet-100 Classes trained and evaluated on as defined by (Du et al., 2024)

iterate between fine-tuning $f$ and generating synthetic data for additional generalization improvements? In this experiment, we generate synthetic data according the original schedules defined in Table 5 for the first 100 epochs. We then continue fine-tuning and generating an additional $1\times$ expansion every 5 epochs for a total of 300 epochs and report generalization performance. We observe that in all cases, we can improve performance, but that gains eventually saturate. It remains a question for future work whether higher-capacity generative models could be mined longer periods of time before predictive performance saturates. If so, it suggests an exciting future possibility of continuously exchanging unused offline compute for improved predictive model performance.

## A.5. Computational Costs

EPISTEMIC HEAD

The Epistemic Head with $K$ heads has a parameter count equal to $K \times d_{\mathrm{model}} \times C$, for model embedding dimension $d_{\mathrm{model}}$ and number of output logits $C$. For many predictive models, this leads to negligible parameter count increase, as displayed in Table 9 – less than 5% increase for all experiments in this paper. Training and inference times are impacted by less than 2% for our most expensive (ImageNet-LT) experiments.

| Classes | Epistemic Heads | |
| --- | --- | --- |
| | 3 | 5 |
| 10 | 0.03% | 0.04% |
| 100 | 0.27% | 0.45% |
| 1000 | 2.67% | 4.44% |

*Table 9.* Epistemic Head Parameter Count as a Percentage of ViT-B Parameter Count

LONGTAIL GUIDANCE

We generate $512 \times 512$ resolution synthetic images at an unoptimized rate of 6.32 image / second for baseline text-prompted diffusion and 1.01 image / second for diffusion with Longtail Guidance (using Stable Diffusion v1.4 in FP16 with 50 DDIM sampling steps on 8xH100 GPUs). The majority of the LTG guidance cost is in differentiably decoding through the VAE. In particular, in the gradient

$$\frac{d\hat{z}_t^0}{dz_t} \times \frac{d\hat{x}_t^0}{d\hat{z}_t^0} \times \frac{df_\phi^{\mathrm{lt}}}{d\hat{x}_t^0} \tag{12}$$

calculated by the main paper's Algorithm 1, the second term is VRAM-intensive (45GB for batch size 8 without gradient checkpointing). This cost occurs primarily because the $83.6m$ parameter VAE decodes latent dimensions of $64 \times 64 \times 4$ to data dimension $512 \times 512 \times 3$.

## A.6. Longtail Signals

In Figure 11 we show that high quantiles $q = 90\%$ of model-based longtail signals are all strong indicators of examples that are disproportionately hard for the model (as determined by average accuracy above and below the quantile). However, as shown in Figure 5 of the main paper, the Epistemic Head more effectively *detects* rare or hard examples.

In Figure 12, we visualize a toy example of high epistemic (left) and high aleatoric (right) uncertainty for three Epistemic Heads each classifying over three classes. High epistemic uncertainty occurs when each sample from the predictive posterior

(i.e. each Epistemic Head) gives mutually incompatible answers (such as all being confident about a different class). High aleatoric uncertainty (which includes high entropy) occurs when each sample has similarly and highly ambiguous (e.g. uniform) belief over classes.

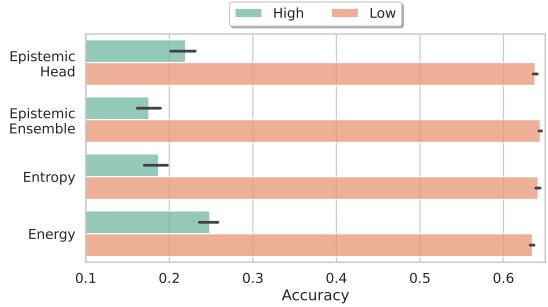

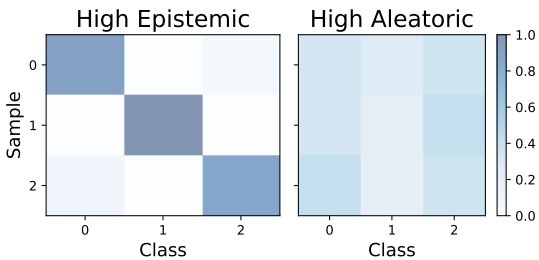

*Figure 11.* High quantiles of longtail signal indicate examples that are disproportionately hard.

*Figure 12.* Probability of classification under high epistemic and high aleatoric uncertainty. There are three samples from the posterior predictive (y-axis). Each maintains a distribution over three classes (x-axis). Epistemic uncertainty (left) goes high when each sample has mutually incompatible beliefs. Aleatoric uncertainty (right) goes high when samples are all highly uncertain.

### A.7. Why Does Longtail Guidance Work without Training the Predictive Model on Intermediate Diffusion States?

A key finding of this work is that an existing predictive model $f_\phi$ does not need to be retrained on intermediate, noisy diffusion states to effectively guide diffusion model $\epsilon_\theta$ towards high-value synthetic training examples that are rare or hard from $f$'s perspective. This realization frees us from the dilemma of having to decide whether to waste predictive model capacity training on intermediate diffusion data it will never see in production or risking divergence from the production model by fine-tuning on intermediate noisy states. But why does this work?

In Figure 13, we visualize the intermediate, noisy diffusion states of two quantities: the decoded, predicted terminal state $D(P(\hat{x}_t^0))$ (top row, what LTG uses as guidance input to predictive model $f_\phi$) and the decoded data state, $D(x_t)$ (bottom row, what classifier guidance would traditionally performs guidance on if not in a latent space). We observe that the terminal state predictions much more readily resemble natural image data at a much earlier time in the diffusion process (within the first 10% of denoising steps) than do the data states. In fact, decoded data states have off-distribution noise artifacts up through the first 90% of the diffusion denoising steps. We speculate that one reason the original classifier guidance work (Dhariwal & Nichol, 2021) performed guidance on intermediate noisy states is that they can be more efficiently generated than the predicted clean terminal states; intermediate states merely require sampling a random clean data point, sampling a random timestep, and applying the noise schedule. Predicted clean terminal states from a given denoising timestep additionally require a forward pass from the denoising network (see Equation 8).

We make this quantitative with the Frechet Inception Distance (FID) (Heusel et al., 2017), defined as

$$\text{FID} = \|\mu_r - \mu_g\|^2 + \text{Tr}(\Sigma_r + \Sigma_g - 2(\Sigma_r \Sigma_g)^{\frac{1}{2}}) \tag{13}$$

where $(\mu_r, \Sigma_r), (\mu_g, \Sigma_g)$ are the mean and covariance of Inception-V3's `pool3` features for real and synthetic data, respectively. Lower FID indicates that the generated data more closely match and cover the real data. We measure FID for two conditions:

1. FID between the decoded, predicted terminal state $\hat{x}_t^0 = D(P(x_t))$ and real ImageNet-LT training data, and

2. FID between the decoded data state $x_t$ and real ImageNet-LT training data.

Results are plotted in Figure 15. Observe that the decoded, predicted terminal state $D(P(\hat{x}_t^0))$ are dramatically closer in distribution to real training data than are the naively decoded data states, $x_t$ for nearly all denoising iterations.

Figure 14 demonstrates that, because the terminal state estimates $D(P(\hat{x}_t^0))$ are closer real training data, the predictive model is able to effectively guide data generations towards higher longtail signals (model entropy in this case), with clear

longtail signal separation between different longtail guidance weights occurring within the first $25\%$ of denoising iterations. In contrast, naively guiding based on data states $x_t$ causes the predictive model to be unable to effectively guide data generation until the last $5\%$ of denoising iterations, when there are no longer many degrees of freedom for the diffusion model to meaningfully change image content.

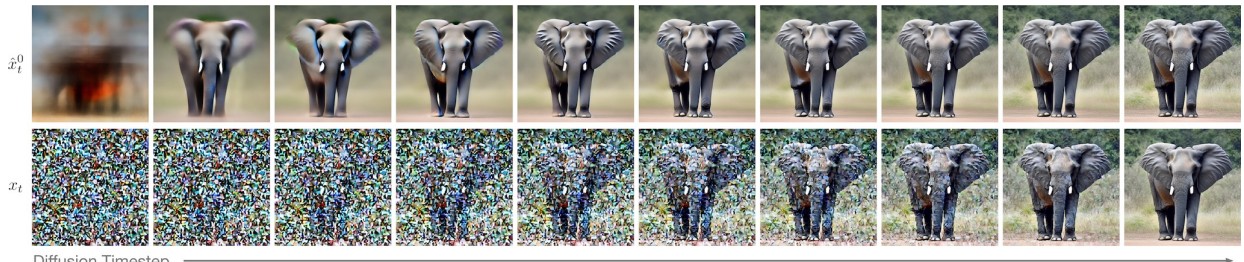

*Figure 13.* Performing Longtail Guidance on predicted, decoded terminal states $\hat{x}_t^0$ (top row) provides production model $f_\phi$ with data that are more in-distribution and less corrupted by intermediate diffusion noise than does Naive guidance performed on each intermediate decoded state $x_t$ (bottom row). See Figures 14, 15 for quantitative analysis.

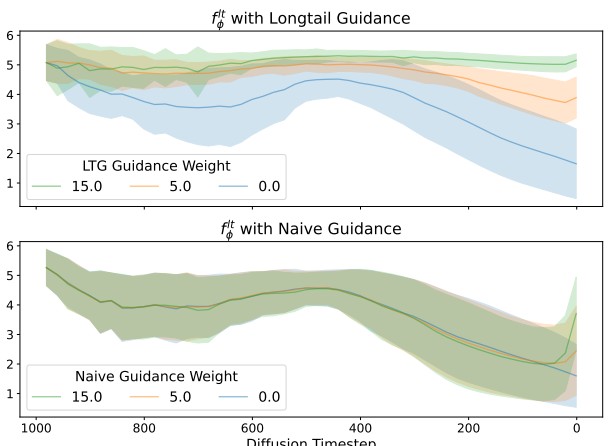

*Figure 14.* Performing Longtail Guidance on the predicted, decoded terminal data state $\hat{x}_t^0$ (top row) provides production model $f_\phi$ with cleaner data much earlier in the diffusion denoising process, enabling it to meaningfully exercise guidance as compared to naively performing guidance on the predicted decoded data state $x_t$. Y-axes are the guiding model longtail signal (in this case, entropy). See Figure 13 for a visual depiction.

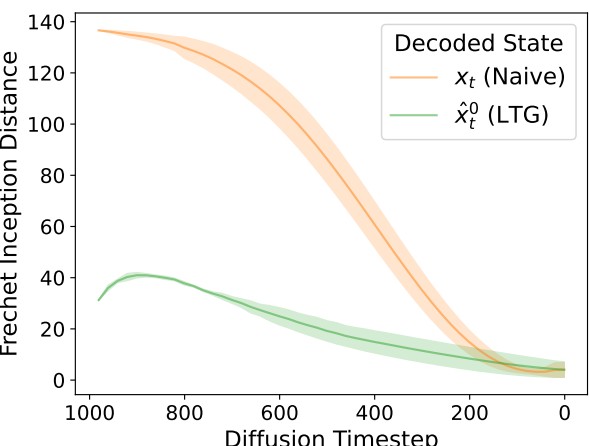

*Figure 15.* Decoded terminal data states $\hat{x}_t^0$ have better FID with respect to real training data than do intermediate data states $x_t$, suggesting that LTG works with an existing predictive model $f_\phi$, even though it is not trained on intermediate diffusion states, precisely because the states upon which LTG performs guidance better match what the predictive model has already seen.

### A.8. Ablations

In Table 16, we ablate over Longtail Guidance signal (energy, entropy, epistemic) by Longtail Guidance Weight (1.0, 10.0, 50.0, 200.0) for the Pets dataset. In Table 17, we ablate over the number of Epistemic Heads on the same task. In both cases, we report predictive model generalization performance when trained on the $30\times$ synthetic data expansion task as defined in Section 4.1. We find highest performance for energy and entropy at guidance weight 10.0 and highest performance for epistemic at guidance weight 50.0. Similarly, we find highest performance for $K = 5$ Epistemic Heads. Performance declines with additional heads, likely because the oracle loss used to train the Epistemic Head causes each head to be exposed to $\frac{1}{K}$ examples in expectation; too many heads and they lose the ability to represent the predictive model.

### A.9. Additional Examples

In Figure 18, we show high-resolution examples of synthetic data generated by baseline diffusion and by Longtail Guidance. Guidance is performed by a SOTA ViT-based ImageNet-LT classifier described in Section 4.2.

| Longtail Signal | Longtail Guidance Weight | | | |
|---|---|---|---|---|
| | **1.0** | **10.0** | **50.0** | **200.0** |
| Energy | 78.8 | **80.0** | 78.9 | 76.7 |
| Entropy | 76.1 | **81.6** | 80.9 | 74.9 |
| Epistemic | 76.8 | 78.5 | **82.0** | 79.1 |

*Figure 16.* Ablation over Longtail Guidance Weight on Predictive Model Generalization Performance on the Pets Dataset Expansion Task, holding the number of Epistemic Heads fixed at $K = 5$. Entries are bolded for best hyperparameter in each row.

| Epistemic Heads | 3 | 5 | 7 | 9 |
|---|---|---|---|---|
| LTG (Epistemic) | 81.2 | **82.0** | 80.6 | 79.9 |

*Figure 17.* Ablation over Epistemic Heads on Predictive Model Generalization Performance on the Pets Dataset Expansion Task, holding the guidance weight fixed at 50.0.

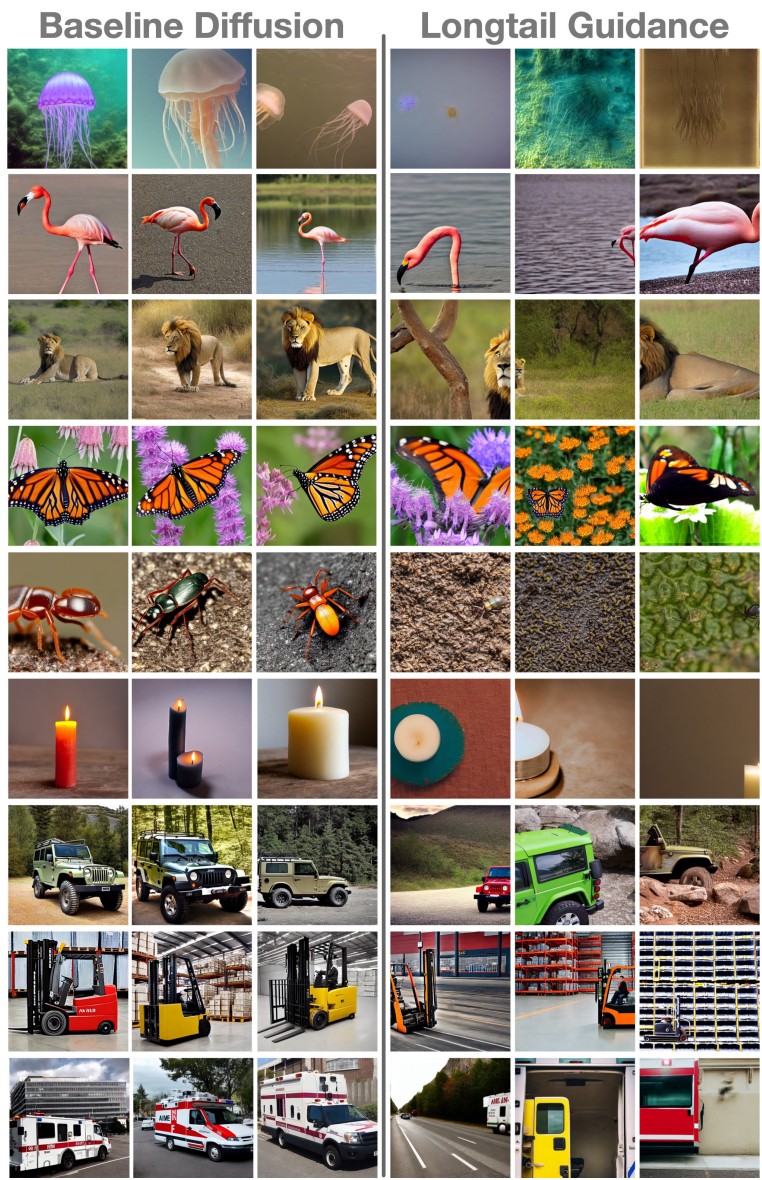

*Figure 18.* Additional example of baseline diffusion (left) and Longtail Guidance (right). Baseline diffusion tends to generate canonical, well-posed views that help a predictive model generalize, but only up to a point. In contrast, Longtail Guidance produces more extreme, occluded, or challenging views from the perspective of a predictive model, enabling significantly improved generalization performance. In order, classes are: jellyfish, flamingo, lion, monarch butterfly, ant, candle, jeep, forklift, and ambulance.

