# OpenReview forum: "Generative Data Mining with Longtail-Guided Diffusion"
_ICML.cc/2025/Conference — ICML 2025 poster_

### Official Review · Reviewer_iGg6 · 2025-03-10

**Overall Recommendation:** 2

**Summary:**

The paper proposes Longtail Guidance (LTG), a method for generating rare or difficult training examples that a predictive model struggles with. The authors propose using an epistemic head to estimate uncertainty and use its signals guide a diffusion model to synthesize hard or rare data without retraining either the predictive or generative model. The authors present experiments showing that LTG-generated data significantly improve model generalization.

**Claims And Evidence:**

Most claims are empirically supported, but some rely on unverified assumptions or could benefit from stronger theoretical grounding.

1. The improved generalization accuracy results ImageNet-LT and other benchmarks are quite broad (including the supplementary results).
2. The epistemic head’s comparison with entropy and energy baselines are fairly convincing.
3. The claims on in-distribution and hard longtail samples are not explicitly verified. The paper assumes that guiding diffusion models using uncertainty signals produces semantically valid longtail examples, but it lacks a principled evaluation of whether these synthetic samples reflect real-world distribution shifts or are just artifacts of the model’s biases.
  a) Is there any principled way to understand whether you can even trust the long-tail data generated? For example, many examples on the mid-to-high LTG look unplausible (e.g. Figure 1, bottom right figures, Figure 6 -- several unplausible images for each class--).
  b) the authors suggest using these examples for retraining. How do you deal with self-reinforcing biases (e.g. given point 3.a)?

**Essential References Not Discussed:**

N/A

**Experimental Designs Or Analyses:**

See prev. points on "Methods And Evaluation Criteria"

**Methods And Evaluation Criteria:**

The methods seem appropriate. Using guiding a diffusion for generating hard training samples is also a sensible choice.

The authors evaluate on several sensible classification tasks and the improvements in rare or difficult examples seem compelling. However, the authors should include additional basic explanatory information on the experimental set-up. In Section 4.1 it is not immediately obvious why there is a need to generate 20x or 30x the amount of training data. The "Dataset Expansion task as defined in GIF (Zhang et al., 2023b)." should be at least minimally explained, and the reader shouldn't need to consult another paper to understand the results in this work. It is not clear what the authors mean by "for parity", which is used often. Similar comments apply for the rest of this section, which we suggest the authors revise.

**Other Comments Or Suggestions:**

--

**Other Strengths And Weaknesses:**

--

**Questions For Authors:**

1. Is there any principled way to understand whether you can even trust the long-tail data generated? For example, many examples on the mid-to-high LTG look unplausible (e.g. Figure 1, bottom right figures, Figure 6 -- several unplausible images for each class--).

2. the authors suggest using these examples for retraining. How do you deal with self-reinforcing biases (e.g. given point 3.a)?

3. see previous comments on "Methods And Evaluation Criteria"

4. see previous comments on "Relation To Broader Scientific Literature"

**Relation To Broader Scientific Literature:**

The prior work section is lacking focus, the work on diffusion isn't overly relevant, the remaining work isn't compared critically to the current work.

The paper’s main idea to use energy signals to guide diffusion sampling are common in the classifier guidance and plug-and-play diffusion literature (e.g., Loss-Guided Diffusion, Diffusion Posterior Sampling). However, the paper does not explicitly contextualize these works.

The finding on the possibility to guide diffusion without explicitly training an inference model on different noise levels is well known in the field. Many papers show that you can use signals from the data space for guidance without the need to learn an inference models at different noise stages, e.g. [1,2], and a substantial corpus of literature on finetuning diffusion models which achieves the same, e.g. [3, 4].

[1] Hyungjin Chung, Jeongsoo Kim, MichaelThompson Mccann, MarcLouisKlasky, andJongChul Ye. Diffusion posterior sampling for general noisy inverse problems. International Conference on Learning Representations (ICLR), 2023.

[2] Jiaming Song, Qinsheng Zhang, Hongxu Yin, Morteza Mardani, Ming-Yu Liu, Jan Kautz, Yongxin Chen, and Arash Vahdat. Loss-guided diffusion models for plug-and-play controllable generation. International Conference on Maching Learning (ICML), 2023."

[3] Venkatraman, S., Jain, M., Scimeca, L., Kim, M., Sendera, M., Hasan, M., Rowe, L., Mittal, S., Lemos, P., Bengio, E., et al. Amortizing intractable inference in diffusion models for vision, language, and control. Neural Information Processing Systems (NeurIPS), 2024

[4] Domingo-Enrich, C., Drozdzal, M., Karrer, B., and Chen, R. T. Q. Adjoint matching: Fine-tuning flow and diffusion generative models with memoryless stochastic optimal control. arXiv preprint arXiv:2409.08861, 2024.

**Theoretical Claims:**

The authors do not present major theoretical results.

---

> ### Author Rebuttal · Authors · 2025-03-31
>
> We thank the reviewers for their time and insights. We are grateful for two solid accepts [Yasy, Qmgq] and believe that we can fully address the concerns of the weak reject [iGg6]. We respond as best as we are able to the extremely brief review of [srKG].
>
> We are pleased to hear from reviewers that “the contributions of this paper are novel and advance the field of longtail data generation in diffusion models,” that our approach of “conditioning generation on model longtail signals is a natural and effective approach” for exposing model weaknesses, that “iterative fine-tuning for generating synthetic data throughout training is a strong design choice, as it adapts to evolving model weaknesses rather than generating all synthetic data at once,” and that “the epistemic head’s comparison with entropy and energy baselines are convincing.” We are also pleased to learn that most reviewers (Qmgq, Yasy, iGg6) agree that our evaluations are sound and compelling.
>
> 1. [igG6] is concerned that the paper’s main idea is diffusion guidance based on energy signals and requests contextualization of additional related works.
>
> Our main idea is grounding the notion of "predictive model longtail" into measurable signals. We demonstrate that we can not only generate additional longtail examples by this definition without changing the predictive model or the diffusion model, but that these additional examples significantly improve predictive model generalization on real eval data. Diffusion guidance plays  a supporting role. We also show that predictive model longtails are model-specific (Figure 9) and proactively explain a predictive model's longtail with human-interpretable text (Table 3).
>
> We agree that existing works have already demonstrated flexible guidance schemes in data space and, in fact, compare against one such work (Universal Guidance) in Section 3 (lines 260-290). We also provide new evidence based on FID scores for why this type of guidance works in Supplemental A.8 (Figures 13-15). We will expand the related works discussion (including the additional references) with the ninth page allowed in the camera ready
>
> Universal Guidance, which we directly compare to in Section 3 and Figure 7, already cites the Diffusion Posterior Sampling paper [1] (and its relationship to it). In short, Universal Guidance and Diffusion Posterior Sampling are related in that both use a point estimate for p(x_0 | x_t) when performing guidance. Diffusion Posterior Sampling is focused on Gaussian and Poisson measurement noise (though their framework extends to nonlinear inverse problems where gradients can be calculated). Universal Guidance represents this extension (through the use of a nonlinear observation model represented by the diffusion model’s associated VAE) and takes it two steps further with expensive backward and recurrent sampling steps discussed in our paper, lines 260-270. Longtail Guidance exists between Diffusion Posterior Sampling and Universal Guidance because it differentiably decodes from a diffusion latent to the data space through a nonlinear, differentiable observation model (the VAE), but does not perform additional sampling steps (which are expensive and cause synthetic data to fall out of distribution, see Figure 7).
>
> Loss-Guided Diffusion [2] builds on [1] by using Monte Carlo samples from an approximating distribution to further improve guidance. We do not use it in our approach since we would have to calculate the gradient through the VAE for each sample, which we already note to be an expensive operation (lines 292-295 and Supplement A.6). We will include and contextualize [1], [2, 3, 4] to our list of related works on diffusion control methods (ControlNet, DOODL, Freedom, Edict) and distinguish by whether or not they require additional diffusion or predictor training.
>
> We emphasize that we are primarily concerned with downstream predictive model performance – a task that neither Diffusion Posterior Sampling nor Loss-Guided Diffusion evaluate on (but which our strongest baselines, GIF, Dream-ID, and LiVT do).
>
>
> 2. [iGg6] is concerned about whether we address model bias, distribution shift, and trustworthiness of synthetic data.
>
> Please see response 5. to [Yasy].
>
>
> 3. [iGg6] asks for clarification and inlined descriptions of our experiment setup.
>
> We fully describe our experiments in Supplement A.2. We will inline in the main paper with the ninth page allotted for the camera ready. Please also note that the 20-30x expansion is only for one apples-to-apples evaluation (with GIF). Other evaluations use less than 1x expansion of synthetic data. See response 6. to [Yasy].
>
>
> 4. [igG6] is concerned that some examples with mid-to-high LTG weight look implausible
>
> The implausibility of some synthetic data with high Longtail Guidance weights are displayed for demonstration purposes only. The guidance weights in the experiments are in the low-to-mid range of the qualitative examples. We will clarify in the camera-ready

---

> > ### Comment · Reviewer_iGg6 · 2025-04-03
> >
> > Thank you for providing answers to the raised concerns.
> >
> > It would be worthwhile to have some guarantees, but it does not seem possible within this framework (point 3.a).
> > Point 3.b was not clearly/explicitly addressed.
> > Most of the raised concerns here still remain. The 20-30x data expansion clarification was useful, thank you.
> > This point is much clearer, thank you.

---

> > > ### Author Response · Authors · 2025-04-06
> > >
> > > We are glad to learn that iGg6 is convinced about dataset sizes, and it appears they are also convinced that our contributions are novel (response 1 to iGg6).
> > >
> > > The remaining concerns seem to be:
> > >
> > > 3.a “is there any principled way to understand whether you can even trust the long-tail data generated,” and
> > >
> > > 3.b “How do you deal with self-reinforcing biases,”
> > >
> > > We have the same answer to each question (which we believe we previously answered in response 5 to Yasy but elaborate on here):
> > >
> > > Concerns 3.a and 3.b are held in check by two sources:
> > >
> > > (i) the predictive model’s own probability estimates, and
> > >
> > > (ii) generalization performance on real evaluation data
> > >
> > > (i): we ensure that the probability of the target class under the predictive model is lower for longtail synthetic data generation than for baseline synthetic data generation while also not approaching zero. Thus, the probability of the target class under the predictive model is itself a guardrail since, if it goes to zero, we have likely drifted OOD. We also show that the model’s own longtail signals increase with longtail guidance weight, demonstrating that longtail-guided synthetic data are, in fact, longtail by our definition. See Figure 2 and lines 238-258.
> > >
> > > (ii): if generalization improves on real evaluation data, then we claim a predictive model is more capable. If the generated synthetic data were OOD or if the predictive model’s own biases caused self-reinforcing or “runaway” processes of biased, low-quality synthetic data then they will be revealed by weaker performance on real evaluation data, at which point we can rewind to a pre-regression checkpoint.
> > >
> > > But, in fact, we see no evidence of regression for any of the low-to-mid longtail guidance weights used in our experiments (Section 4.1, Supplement A.1, Tables 1, 2, 4), which consist of eight datasets, as many as 1000 classes, all with substantial gains on generalization performance over strong data augmentation, adversarial perturbation, synthetic data, and longtail mitigation baselines – even when training continues for hundreds of epochs (Supplement A.5, Table 8).
> > >
> > > We further address concerns 3.a, 3.b by showing in Sec. 4.2 that VLM descriptions of longtail synthetic data outperform VLM descriptions of baseline synthetic data when those descriptions are used to generate new synthetic training data.
> > >
> > > While trust and bias are valid concerns, it is also a concern that synthetic data generated without longtail signals from the predictive model more rapidly saturate in new concepts (see response 6 to [Yasy]), leading to weaker generalization. Our experiments strongly support that giving the predictive model “a voice” in the generating process is to its benefit.
> > >
> > > If the reviewer is not convinced by the guardrails (i) and (ii) we already use, then may we ask what would be convincing? It seems that it cannot be signals from another predictive model since that model could also be biased. And it cannot be other real data, since that data could also be folded into additional evaluation data.
> > >
> > > Finally, we provide additional evidence that longtail synthetic data are, class for class, distributionally closer to real training data than real training data are to themselves (across class boundaries). We compute evaluation metrics FID [1] and generative precision+recall [2] on the fine-grained Flowers dataset between the following distributions:
> > >
> > > 1. real_real: measured between all pairs of real training data classes
> > >
> > > 2. real_baseline: measured between each class of real training data and the corresponding class of baseline synthetic data (no longtail guidance)
> > >
> > > 3. real_longtail: measured between each class of real training data and the corresponding class of longtail-guided synthetic data
> > >
> > > 4. real_noise: measured between each class of real training data and a set of images containing pure Gaussian noise.
> > >
> > > We find:
> > >
> > > Recall (likelihood of distribution1 samples against the distribution2 manifold, higher is better)
> > >
> > > real_noise < real_real < real_baseline < real_longtail
> > >
> > > Precision (likelihood of distribution2 samples against the distribution1 manifold, higher is better)
> > >
> > > real_noise < real_real < real_longtail < real_baseline
> > >
> > > FID (lower is better)
> > >
> > > real_noise > real_real > real_longtail > real_baseline
> > >
> > > Longtail synthetic data have better FID, precision, and recall wrt real data (comparing within same classes) than real data has to itself (comparing between classes). Thus, longtail data not only remain in-distribution, but they remain concentrated within class boundaries. Baseline synthetic data have higher precision and lower FID than longtail synthetic data, supporting that longtail guidance generates samples that are further from the mode of real data manifold (as we would expect), while longtail synthetic data have higher recall than baseline synthetic data, supporting that longtail synthetic data are more diverse than baseline synthetic data.
> > >
> > > [1] Heusel et al., NeurIPS ‘17
> > >
> > > [2] Kynkäänniemi et al., NeurIPS ‘19

---

### Official Review · Reviewer_Yasy · 2025-03-13

**Overall Recommendation:** 3

**Summary:**

The paper proposes a diffusion model guidance technique (LTG) that generates synthetic training data tailored to the specific long-tail of a deployed predictive model. It introduces a differentiable module for estimating epistemic uncertainty that helps identify rare or hard examples without altering model weights or predictive performance. It also proposes method that extracts textual descriptions of model weaknesses from LTG-generated data using vision-language models (VLMs).

**Claims And Evidence:**

The paper claims LTG does not require retraining of the diffusion or predictive model; would retraining the predictive model on intermediate diffusion states further improve performance?
While the generalization improvements suggest that LTG-generated data are useful, the paper does not quantitatively verify whether the generated data truly remain in-distribution as claimed.

**Essential References Not Discussed:**

N/A

**Experimental Designs Or Analyses:**

The experimental design and analyses in the paper appear generally sound. The iterative fine-tuning approach for generating synthetic data throughout training is a strong design choice, as it adapts to evolving model weaknesses rather than generating all synthetic data at once.

**Methods And Evaluation Criteria:**

LTG is designed to expose model weaknesses, so conditioning generation on model longtail signals is a natural and effective approach.
The benchmark datasets include real-world longtail scenarios, and tested on imbalanced, fine-grained, and out-of-distribution datasets.
The central problem in the use of diffusion model for synthesising images useful for training classifiers is the sheer amount of synthetic data needed for meaningful classifier improvements. The paper doesn't quite address this problem. The method still generates synthetic data at a scale of 20× to 30× the original dataset size.

**Other Comments Or Suggestions:**

N/A

**Other Strengths And Weaknesses:**

N/A

**Questions For Authors:**

N/A

**Relation To Broader Scientific Literature:**

The paper nicely incorporates ideas from deep learning-based uncertainty estimation literature to address an important issue in using diffusion models for generating synthetic data. It would be interesting to explore how well this method generalizes to more recent diffusion models and to evaluate its data efficiency, given that it still generates data at >= 30 times the size of the existing dataset.

**Theoretical Claims:**

N/A

---

> ### Author Rebuttal · Authors · 2025-03-31
>
> We thank the reviewers for their time and insights. We are grateful for two solid accepts [Yasy, Qmgq] and believe that we can fully address the concerns of the weak reject [iGg6]. We respond as best as we are able to the extremely brief review of [srKG].
>
> We are pleased to hear from reviewers that “the contributions of this paper are novel and advance the field of longtail data generation in diffusion models,” that our approach of “conditioning generation on model longtail signals is a natural and effective approach” for exposing model weaknesses, that “iterative fine-tuning for generating synthetic data throughout training is a strong design choice, as it adapts to evolving model weaknesses rather than generating all synthetic data at once,” and that “the epistemic head’s comparison with entropy and energy baselines are convincing.” We are also pleased to learn that most reviewers (Qmgq, Yasy, iGg6) agree that our evaluations are sound and compelling.
>
>
> 5. [Yasy] says, “the paper does not quantitatively verify whether the generated data truly remain in-distribution.”
>
> Thank you for raising this concern. Longtail guidance demonstrates significant generalization improvements over strong synthetic data generation baselines across eight datasets, with as many as 1000 distinct classes (Tables 1, 2, Supplemental A.1). It is common practice to trust (real) evaluation data and, in fact, our method enables users to reserve more real data for evaluation by more heavily relying on synthetic training data.
>
> Furthermore, in Section 4.2 and Table 3, we show that forming new diffusion prompts from VLM descriptions of longtail-guided synthetic data and then training the original predictive model on that new data outperforms the baseline where diffusion prompts are formed from VLM descriptions of baseline synthetic data (without longtail guidance). This quantitatively supports that longtail-guided data are meaningful and remain in-distribution.
>
> We agree that there is a tension between synthetic data distributional alignment and predictive model bias. Because we are working with natural image datasets, alignment can be equated with realism. Alignment and realism are explicitly controlled by the diffusion model’s text guidance weight. Predictive model bias is explicitly controlled by the longtail guidance weight.
>
> We address the tradeoff between alignment and bias by holding the text guidance weight constant and selecting a longtail guidance weight such that the probability of the desired class (under the current predictive model) is lower than baseline synthetic data but not so low that it drops to zero. It is a hyperparameter that was selected one time – it does not need to be done for each generation or dataset. We describe this in lines 238-258 and show in Figure 2 that there is a strong inverse relationship between longtail guidance weight and the probability of correct classification under the predictive model for generated data (before retraining).
>
> More broadly, while predictive model bias is a concern, it is also a concern that synthetic data generated without signals from the predictive model more rapidly saturate in new concepts (see rebuttal 6 to [Yasy]), leading to weaker generalization improvements. Our experiments strongly support that giving the predictive model “a voice” in the generating process is to its benefit.
>
>
>
> 6. [Yasy] is concerned that, “the central problem in the use of diffusion models for synthesising images useful for training classifiers is the sheer amount of synthetic data needed for meaningful classifier improvements. The paper doesn't quite address this problem. The method still generates synthetic data at a scale of 20× to 30× the original dataset size.”
>
> We note that the results in Table 2 (Dream-ID comparison) all generate longtail-guided data at less than 1x the size of the training data and that, in Supplemental A.1, we show ImageNet-LT results that generates much fewer synthetic data – just 20% of the original dataset and, yet, still yield significant generalization improvements!
>
> The 20-30x expansion in Table 1 is for comparing apples-to-apples with the GIF baseline (this is what we mean by “for parity.”). Further inspection of our existing results shows that predictive model generalization from our longtail guidance method surpasses GIF in all cases using just 20-30% (average 25%) of the data expansion that GIF uses (caltech: 4x expansion vs 20x), cars (5x expansion vs 20x), pets (7x expansion vs 30x), flowers (6x expansion vs 20x). Training with even more longtail-guided data yields further generalization improvements as we report in the paper.
>
> We reported the equivalent expansion numbers in the paper for direct comparison, but we will take advantage of the ninth page in the camera ready to also add a graph that demonstrates that longtail guidance drives generalization improvements in a much more data-efficient way (4x more efficient) compared to GIF!

---

### Official Review · Reviewer_Qmgq · 2025-03-14

**Overall Recommendation:** 4

**Summary:**

This paper proposes a novel approach to generate long-tail data using diffusion models. The authors introduce an epistemic head and a long-tail guidance mechanism, enabling the model to detect and generate long-tail data effectively. Experimental results demonstrate that the proposed Longtail-Guided Diffusion model significantly enhances dataset quality, as evidenced by improved downstream task performance and meaningful data generation.

**Claims And Evidence:**

The claims presented in the paper are clear and well-articulated.

**Essential References Not Discussed:**

No essential references appear to be missing from the discussion.

**Experimental Designs Or Analyses:**

The experimental designs and analyses are sound and support the claims made in the paper.

**Methods And Evaluation Criteria:**

The proposed methods and evaluation criteria are well-founded and appropriate.

**Other Comments Or Suggestions:**

N/A.

**Other Strengths And Weaknesses:**

The strengths and weaknesses of the paper have been thoroughly addressed in the sections above.

**Questions For Authors:**

Could the authors provide computational efficiency about applying the longtail-guiided diffusion?

**Relation To Broader Scientific Literature:**

The contributions of this paper are novel and advance the field of longtail data generation in diffusion models.

**Theoretical Claims:**

The theoretical claims and proofs presented in the paper are sound and well-justified.

---

> ### Author Rebuttal · Authors · 2025-03-31
>
> We thank the reviewers for their time and insights. We are grateful for two solid accepts [Yasy, Qmgq] and believe that we can fully address the concerns of the weak reject [iGg6]. We respond as best as we are able to the extremely brief review of [srKG].
>
> We are pleased to hear from reviewers that “the contributions of this paper are novel and advance the field of longtail data generation in diffusion models,” that our approach of “conditioning generation on model longtail signals is a natural and effective approach” for exposing model weaknesses, that “iterative fine-tuning for generating synthetic data throughout training is a strong design choice, as it adapts to evolving model weaknesses rather than generating all synthetic data at once,” and that “the epistemic head’s comparison with entropy and energy baselines are convincing.” We are also pleased to learn that most reviewers (Qmgq, Yasy, iGg6) agree that our evaluations are sound and compelling.
>
> 7. [Qmgq] asks about computational efficiency
>
> Computational efficiency is discussed for Longtail Guidance and the Epistemic Head in Supplement A.6. In brief, we generate baseline synthetic images (no Longtail Guidance) at a rate of 6.32 images / second (50 DDIM steps, fp16, no gradient checkpointing, 8xH100). We generate Longtail-Guided synthetic images at 1.01 images / second. The largest cost is the gradient calculation through the VAE, which consumes 45GB of VRAM in fp16 for batch size 8. The Epistemic Head impacts training and inference times by less than 2% and contains less than 5% of the original predictive model’s parameters. See also our response about 4x increased data efficiency over GIF in Response 6 to Yasy.

---

### Official Review · Reviewer_srKG · 2025-03-14

**Overall Recommendation:** 1

**Summary:**

This paper proposes a proactive long-tail discovery process that helps the model learn rare or hard concepts.
Specifically, the authors develop model-based long-tail signals and use these signals to generate additional training data from latent diffusion models.

**Claims And Evidence:**

Producing rare or hard concepts is also an important issue in generative tasks. Applying this to a classification task, which is easier than a generative task, is like using a sledgehammer to crack a nut.
Indeed, the dataset used in the experiments is much smaller than the one on which the Stable Diffusion model is trained.

**Essential References Not Discussed:**

Potential related work:

Um, Soobin, and Jong Chul Ye, Self-guided generation of minority samples using diffusion models, ECCV 2024.

**Experimental Designs Or Analyses:**

If the authors want to validate the scenario in practice, they need to employ a diffusion model trained on a small dataset and use it to train a predictive model for a much larger dataset.

**Methods And Evaluation Criteria:**

.

**Other Comments Or Suggestions:**

.

**Other Strengths And Weaknesses:**

.

**Questions For Authors:**

.

**Relation To Broader Scientific Literature:**

.

**Theoretical Claims:**

.

---

> ### Author Rebuttal · Authors · 2025-03-31
>
> We thank the reviewers for their time and insights. We are grateful for two solid accepts [Yasy, Qmgq] and believe that we can fully address the concerns of the weak reject [iGg6]. We respond as best as we are able to the extremely brief review of [srKG].
>
> We are pleased to hear from reviewers that “the contributions of this paper are novel and advance the field of longtail data generation in diffusion models,” that our approach of “conditioning generation on model longtail signals is a natural and effective approach” for exposing model weaknesses, that “iterative fine-tuning for generating synthetic data throughout training is a strong design choice, as it adapts to evolving model weaknesses rather than generating all synthetic data at once,” and that “the epistemic head’s comparison with entropy and energy baselines are convincing.” We are also pleased to learn that most reviewers (Qmgq, Yasy, iGg6) agree that our evaluations are sound and compelling.
>
> 8. [srKG] states, “if the authors want to validate the scenario in practice, they need to employ a diffusion model trained on a small dataset and use it to train a predictive model for a much larger dataset,” and, “applying this [production of rare or hard concepts] to a classification task, which is easier than a generative task, is like using a sledgehammer to crack a nut.
>
> It is frequently the case that deployed predictive models have access to much less capacity and training data than do larger foundation models (including Internet-scale generative models). This can be due to limited memory and compute budgets and limited opportunities for real data collection (as we note in lines 38-48, 69-73). This is the premise behind the substantial literature of model and data distillation, including the works we cite in the paper (Yu et al., 2023b; Gou et al., 2021), and the more nascent literature on synthetic training data, including the works we cite in the paper (Du et al., 2024; Azizi et al., 2023; Zhou et al., 2023, Zhang et al., 2023b; Li et al., 2022b).
>
> When working with a deployed predictive model, it is critical to understand scenarios that the model struggles with. As we detail in Response 1 to [igG6], our primary contribution is towards defining, mitigating, and proactively understanding a given predictive model’s longtail. Diffusion guidance only plays a supporting role. Thus, while we appreciate the reference on minority diffusion sampling, and will include it in our camera-ready, we also note in our analysis (Section 4.3, Figure 9) that what is difficult for one model (even a foundation model like CLIP) is not necessarily the same thing as what is difficult for a given, deployed predictive model. We in fact show that Longtail Guidance demonstrates significant generalization improvements over strong synthetic data generation baselines (that use a foundation model for diffusion guidance) across eight datasets, with as many as 1000 distinct classes (Tables 1, 2, Supplemental A.1).

---

### Decision · Program_Chairs · 2025-05-01

**Decision:**

Accept (poster)

**Comment:**

The paper presents a guidance technique for diffusion models, called LTG, which generates synthetic training data tailored to the long-tail distribution specific to a deployed predictive model. It introduces a differentiable module for estimating epistemic uncertainty, enabling the identification of rare or challenging examples without modifying model weights or affecting predictive performance. Additionally, it proposes a method to extract textual descriptions of model weaknesses from LTG-generated data using vision-language models (VLMs).

While most reviewers find the result interesting and the studied problem important, the work can be improved by comparing the results more comprehensively and properly discussing the related work. We recommend that the authors incorporate the reviewers' feedback in future versions.